



# Impacts of the COVID-19 lockdown on air pollution at regional and urban background sites in northern Italy.

Jean-Philippe Putaud[1], Luca Pozzoli[1], Enrico Pisoni[1], Sebastiao Martins Dos Santos[1], Friedrich Lagler[1], Guido Lanzani[2], Umberto Dal Santo[2], Augustin Colette[3]

[1]European Commission, Joint Research Centre (JRC), Ispra, Italy
[2]Agenzia Regionale per la Protezione dell'Ambiente (ARPA-Lombardia), Milan, Italy
[3]Institut National de l'Environnement Industriel et des Risques (INERIS), Verneuil-en-Halatte, France

Correspondence to: J.P. Putaud (jean.putaud@ec.europa.eu)

**Abstract**

The COVID-19 lockdown measures gradually implemented in Lombardy (northern Italy) from 23 February 2020 led to a downturn in several economic sectors with possible impacts on air quality. Several communications claimed in the first weeks of March 2020 that the mitigation in air pollution observed at that time was actually related to these lockdown measures, without considering that seasonal variations in emissions and meteorology also influence air quality. To determine the specific impact of lockdown measures on air quality in northern Italy, we compared observations from the European Commission atmospheric observatory of Ispra (regional background) and from the regional environmental protection agency (ARPA) air monitoring stations in the Milan conurbation (urban background) with expected values for these observations using two different approaches. On the one hand, intensive aerosol variables determined from specific aerosol characterisation observations performed in Ispra were compared to their 3-year averages. On the other hand, measured concentrations of atmospheric pollutants ($NO_2$, $PM_{10}$, $O_3$, $NO$, $SO_2$) were compared to expected concentrations derived from the Copernicus Atmosphere Monitoring Service Regional (CAMS) ensemble model forecasts, which did not account for lockdown measures. From these comparisons, we show that $NO_2$ concentrations decreased as a consequence of the lockdown by -30% and -40% on average at the urban and regional background sites, respectively. Unlike $NO_2$, $PM_{10}$ concentrations were not significantly affected by lockdown measures. This could be due to any decreases in $PM_{10}$ (and $PM_{10}$ precursors) emissions from traffic being compensated for by increases in emissions from domestic heating and/or from changes in the secondary aerosol formation regime resulting from the lockdown measures. The implementation of the lockdown measures also led to an increase in the highest $O_3$ concentrations at both the urban and regional background sites resulting from reduced titration of $O_3$ by $NO$. The relaxation of the lockdown measures beginning in May resulted in "close to expected" $NO_2$ concentrations in the urban background, and to significant increases in $PM_{10}$ in comparison to expected concentrations at both regional and urban background sites.



## 1 Introduction

The COVID-19 pandemic is an epidemic of coronavirus disease 2019 (COVID-19), of which the outbreak was first identified in Wuhan, China, in late December 2019. The World Health Organization declared COVID-19 a pandemic on 11 March 2020. The first case of COVID-19 in northern Italy was detected on 20 February 2020 in Codogno, about 60 Km south east of Milan (Figure 1). To reduce the virus spreading, the Italian Government quickly adopted a series of measures, such as the quarantine for 10 municipalities, the cancellation of all main public events and the closure of schools and universities in northern Italy (DL, 23 February 2020). The lockdown started in all of Italy on 9 March 2020 (DPCM, 8 March 2020). All commercial and retail activities were closed on 11 March, except for grocery shops and pharmacies (DPMCM, 11 March 2020) and it was forbidden to move outside the place of residence, except for health issues or work. Further lockdown measures were decreed on 22 March 2020 (DPCM, 22 March 2020), including the suspension of all non-essential industrial production activities. The lockdown lasted until 4 May 2020 (DPCM, 26 April 2020), when a gradual relaxation of the measures was decided by the government. The re-opening of manufacturing industries and construction sites was allowed, but schools and universities as well as some commercial activities such as restaurants remained closed. Movements from a region to another were still forbidden, but moving short distances to work and to visit relatives was possible. From 18 May 2020, most commercial businesses could re-open and free movement was granted within regional borders (DL, 16 May 2020). This lockdown provided a unique opportunity to determine how such dramatic measures can eventually influence air quality. This is the focus of this paper.

Lombardy, Piedmont and Emilia-Romagna in northern Italy produce roughly 50% of the national Gross Domestic Product (GDP), with Lombardy alone producing 22% of the national GDP (Istat, 2018 data). This economic dynamism (mainly linked to industrial production and service related activities) is associated with significant pollutant emissions, which together with unfavourable conditions for pollution dispersion (due to low wind speeds and particular orography) cause high pollution levels leading to exceedances of the EU standards for nitrogen dioxide ($NO_2$), particulate matter ($PM_{10}$) and ozone ($O_3$) in northern Italy (EEA, 2019). In this area, the impact of lockdown on economic activities were quite important, as illustrated by data relative to the production of electricity and energy for heating, and to transport related activities (ARPA Lombardia, 2020). Compared to 2019, the Italian thermal electricity production (Figure 2) fell in March (-18%), April (-24%) and May 2020 (-16%). The consumption of natural gas by the industrial sector as reported by the Italian natural gas provider (www.snam.it) also fell by roughly -30% at the end of March in comparison to the beginning of March 2020.

Regarding transport, the Monitoring of Polluting Vehicles project (MOVE-IN) managed by the Lombardy region provided data on the traffic changes derived from its monitoring of 'vehicle km' driven by light duty vehicle and passenger cars (for a small number of vehicles compared to the full fleet circulating in the region though). MOVE-IN data show that the number of 'vehicle km' driven by light duty vehicles remained quite constant till 9 March 2020, then dropped by -75% to reach a minimum between 16 March and 13 of April 2020 before returning to 'usual' (i.e. as before the lockdown period) values after 4 May 2020 (ARPA Lombardia, 2020). For private cars, the number of 'vehicle km' driven also decreased by roughly -70% between the beginning and the end of March, and started increasing again after the 4th of May, but with a slower recovery than for light duty vehicles. The number of requests for driving directions (www.apple.com/covid19/mobility) showed similar variations (Figure 2).





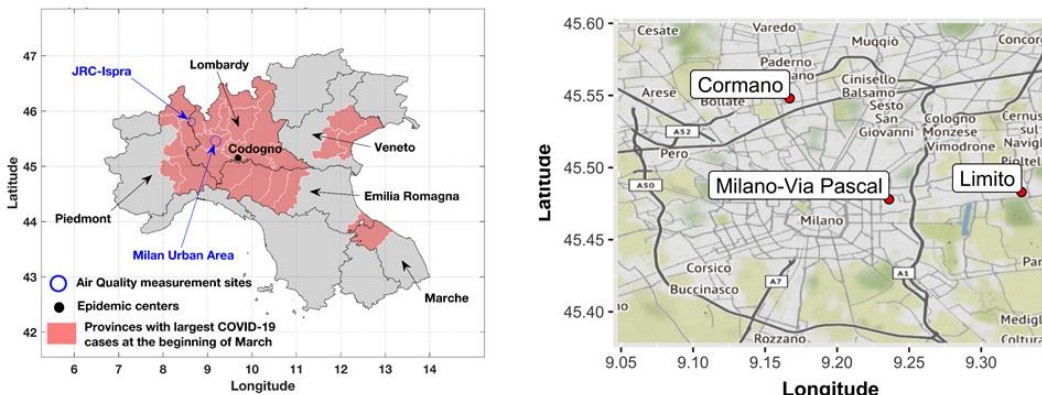

*Figure 1: northern Italy areas impacted by the COVID-19 at the beginning of March 2020, and location of the air pollution measurement sites in Ispra and Milan considered in this study. Right hand panel: © OpenStreetMap contributors 2020. Distributed under a Creative Commons BY-SA License.*

Numerous early communications based on preliminary measurement data analyses associated observed improvements in air quality to the lockdown measures taken to contain the spread of the COVID-19 epidemic. In Brazil, the lockdown in Sao Paolo was followed by drastic reductions in NO (up to -77%) and $NO_2$ (up to -54%) and by an increase in $O_3$ (approximately + 30%) compared to the previous five-year means for the same period (Nakada et al., 2020). In the Yangtze River Delta Region (China), Li et al. (2020) showed that concentrations of $PM_{2.5}$, $NO_2$ and $SO_2$ decreased by 32%, 45% and 20% during a first lockdown phase,

and by 33%, 27% and 7% during a second lockdown phase, compared with the 2017-2019 average for the same period. $O_3$ also increased in that region. In France, the analysis by INERIS (Institut national de l'environnement industriel et des risques) compared air pollution forecast data (calculated without incorporating changes in emissions due to lockdown measures) with adjusted simulations performed *a posteriori* by assimilating observation data influenced by the lockdown measures. They estimated that $NO_2$ concentrations were on average approximately 50% lower than expected in France's largest cities (INERIS, 2020).

Regarding Italy, maps of $NO_2$ surface concentrations estimated from satellite data (e.g. Sentinel-5p) were published by several web sites and media showing large reductions of $NO_2$ concentrations over northern Italy in March 2020 as compared to the previous months and to March 2019 (e.g. Copernicus Atmosphere Monitoring Service headlines published on 17 and 26 of March 2020). Observations and models were also combined in the analysis from the German Aerospace Centre (DLR) which estimated a decrease of about 40% in the total column-integrated $NO_2$ tropospheric concentrations over norther Italy due to the lockdown

measures, using Sentinel-5p data. They also estimated reductions in ground level $NO_2$ concentrations of about -20 µg m$^{-3}$ (-45%) by comparing ground base observations from 25 stations in Lombardy to a model simulation with pre-lockdown emission levels (DLR, 05/05/2020). In-situ observations also showed reduced ground level $NO_2$ concentrations as lockdown measures were implemented. The Environmental Protection Agency ARPA Lombardia showed that March 2020 $NO_2$ concentrations were below the standard deviation calculated from previous years, indicating a possible signal of reduced emissions from traffic and economic

sectors (ARPA Lombardia, 2020). The European Environment Agency (EEA) developed a viewer that tracks $NO_2$ and particulate matter ($PM_{10}$ and $PM_{2.5}$) weekly average concentrations (https://www.eea.europa.eu/themes/air/air-quality-and-covid19). It shows that $NO_2$ concentrations in Milan were at least 24 % lower after the lockdown implementation than during previous weeks, and 21% lower compared to the same period in 2019. Similar trends were found in other cities of northern Italy and European



countries where strong measures were taken to contain the epidemic. In contrast, no consistent effect of the lockdown measures

on particulate matter ($PM_{2.5}$ and $PM_{10}$) could be observed in the main European cities (EEA, 2020).

Air pollution did decline in northern Italy from February to May in 2020 as it does every year, mainly due to seasonal variations in emissions and weather conditions. The strength of certain sources does indeed change during the course of the year, like e.g. domestic heating, while weather conditions influence pollution concentration in diverse ways: advection and dispersion of pollutants resulting from horizontal winds, dilution of pollutants throughout the mixed boundary layer resulting from convection,

and pollutant lifetimes resulting from photochemical reactions (sun radiation), wet removal (clouds and rain), etc. It is therefore not straightforward to disentangle the effects of changing emissions due to lockdown measure implementation from those of seasonal changes in emissions and variability in meteorological conditions between different seasons and different years. In the present study we determine how much of the changes in air pollution observed during the lockdown period in northern Italy were actually due to lockdown measures, independently from expected variations in pollutants' emissions, lifetime and

dispersion. Our results are based on comparisons between air pollution observation data from Ispra (regional background site) and the Milan conurbation (urban background sites) with CAMS-Ensemble model forecast data for the same sites. To help understand the effect of the lockdown measures in regional background area, we also use 4 years of specific aerosol measurements from Ispra.

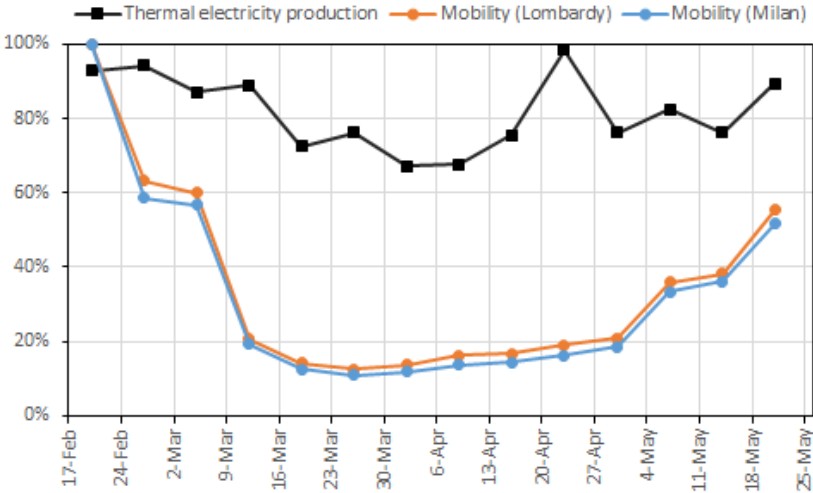

*Figure 2: Variations in activities resulting from lockdown measures (2020). Percentages are calculated in comparison with 2019 data for thermal energy production (source: www.terna.it) and in comparison with data from the third week of February 2020 for mobility data (source: www.apple.com/covid19/mobility).*





## 2 Material and methods

Model and observation air pollution data from 4 sites located in Lombardy covering the time periods 17 February – 24 May 2019 and 2020 were collected and analysed. We selected three sites located in the Milan conurbation as representative of the Milan urban background, and the site of Ispra as representative of the regional background of the upper Po Valley (Figure 1). Ground level concentrations of NO, $NO_2$, $SO_2$, $O_3$ and $PM_{10}$ as measured in-situ at the monitoring stations and as calculated by the CAMS ensemble model forecast were considered. Particle number size distribution and aerosol light absorption Ångström

exponent data from Ispra for the 2017-2020 period were also utilised.

### 2.1 Site description

    The European Commission Atmospheric Observatory (ECAtmO) has been operated in Ispra (45.815 N, 8.636 E, 209 m a.s.l.) since November 1985. It has contributed to the CLRTAP-EMEP (co-operative programme for monitoring and evaluation of the long-range transmission of air pollutants in Europe under the Convention on Long-range Transboundary Air Pollution) and WMO-

GAW (World Meteorological Organization – Global Atmosphere Watch) air pollution measurement programmes for several decades and to the European Research Infrastructures ICOS (Integrated Carbon Observation System) and ACTRIS (Research Infrastructure for the observation of Aerosol, Clouds and Trace Gases) for several years. ECAtmO is located on the northwestern edge of the Po Valley, 20 - 60 km away from major pollution point sources, but still in a densely populated area (ca 500 / km²) with significant economic activity (GDP per capita = 29000 €; EUROSTAT, 2017). Wood burning for domestic heating is also an

important source of particulate matter during the cold period of the year (Gilardoni et al., 2011). Past measurements of $HCHO/NO_2$ ratios compared to the threshold values proposed by Tonnensen and Dennis (2000) suggest that the photochemical production of $O_3$ is limited by the availability of volatile organic compounds in February – May in Ispra.

    The Milan metropolitan area is the second largest densely populated area in Italy (ca 2300 / km²), with a GDP per capita of about 54000 € (EUROSTAT, 2017), and about 4100 circulating vehicles/km² (ISTAT, 2018). Three stations in the Milan conurbation

were selected as representative of the urban background in Milan city, namely 'Milan via Pascal' (45.478 N, 9.236 E, 122 m a.s.l.),' Cormano' (45.548 N, 9.167 E, 155 m a.s.l.), and 'Limito di Pioltello' (45.483 N, 9.328 E, 123 m a.s.l.). All three stations are operated by ARPA Lombardia. We selected only urban background stations because pollutant concentrations at traffic sites are hardly reproducible by regional air quality models with a horizontal resolution of about 10 km. The station in Milan via Pascal is located near the university and it is considered as the urban background station of the city, while the other two stations are located in

the hinterland, near (< 500 m) two main roadways used by commuters at the northern (Cormano, with about 75000 vehicles/day) and eastern (Limito, 20000 vehicles/day) entrances of the city. Average population densities are 7500 / km², 4500 / km², and 2800 / km² in Milan, in Cormano, and Limito di Pioltello, respectively (ISTAT, 2018).

### 2.2 Measurements

    At ECAtmO in Ispra, on-line in situ air pollution measurements are performed from appropriate inlets located at 6.5 and

9 m above ground level for gaseous and particulate pollutants, respectively. The inlet for reactive gas is made of PTFE (i.d. 2.7 cm). The sample residence time in the inlet tube is ca. 2 s. Each analyser samples from the main inlet through a Nafion dryer. In 2019 – 2020, the measurement programme included CO, NO, NOx, $NO_2$, $SO_2$, $O_3$, non-methane hydrocarbons (until 6 March 2020) and $NH_3$ (since 28 January 2020) as gaseous pollutants. The NOx (i.e. NO and $NO_2$), $SO_2$, and $O_3$ data reported in this work were obtained with trace level instruments based on IR (1200 nm) chemiluminescence and Mo converter (ThermoFisher 42iTL), UV





(214 nm) fluorescence (ThermoFisher 43i TLE), and UV (254 nm) absorption (ThermoFisher 49C), respectively. These instruments are calibrated every 3 months using zero air and certified gas cylinders (NO and $SO_2$) or a primary standard ozone generator ($O_3$). In 2019, annual average concentrations of NO, $NO_2$, $SO_2$, $O_3$ and $PM_{10}$ were 4, 16, 0.4, 38 and 21 µg m$^{-3}$, respectively. Particulate matter is sampled through metal-made inlets characterized by negligible losses. Each instrument samples isokinetically from the main aerosol inlet through Nafion dryers. In 2019 – 2020, the aerosol on-line in situ measurement programme included $PM_{10}$

mass concentration, particle number concentration and number size distribution, particle light extinction, absorption, scattering and backscattering at several wavelengths. The $PM_{10}$ mass, particle number and light absorption data reported in this work were obtained with a TEOM-FDMS (ThermoFisher 1405-DF), a Differential Mobility Particle Sizer (Home-made Vienna-type Differential Mobility Analyzer + TSI 3772 Condensation Particle Counter) covering the particle mobility diameter range 10 – 800 nm, and a 7-wavelength Aethalometer (Magee AE31), respectively. The TEOM has been calibrated using a standard filter provided by the

manufacturer, while the DMPS and the Aethalometer are operated, maintained and controlled according to ACTRIS guidelines (www.actris.eu). They were both calibrated at the specific ACTRIS central facility (www.actris-ecac.eu) on 3-7 June 2019. Near real time data are available from the JRC data catalogue at data.jrc.ec.europa.eu/collection/abcis.

The three stations in the Milan conurbation are part of the ARPA Lombardia air quality network, compliant with Directive 2008/50/EC requirements in terms of measurement methods, macro and micro localisation, and data coverage. Inlets are located

at 2.5 m above ground level for all pollutants. The measurement programmes comprise NO, NOx, $NO_2$, $SO_2$, and $O_3$ at all three sites. Additional measurements include benzene, toluene, xylenes, $PM_{10}$, $PM_{2.5}$, B(a)P and $NH_3$ in Milan, and CO and $PM_{10}$ in Limito. Each gas analyser samples from the main inlet through a Nafion dryer. The NOx data reported in this work for the Milan conurbation were obtained with trace level instruments based on IR (1200 nm) chemiluminescence and Mo converter (Teledyne API 201E, ThermoFischer 42 in and ThermoFischer 42c in in Milano Pascal, Limito and Cormano, respectively) and $O_3$ was

measured by UV (254 nm) fluorescence (ThermoFischer 49i at all 3 sites). All measurements are performed according to a specific QA/QC programme. All gas monitors are calibrated every 3 months using zero air and certified gas cylinders (NO) and every 6 months using a primary standard ozone generator for $O_3$. The $PM_{10}$ mass concentrations in the Milan conurbation reported in this work were measured using beta absorption analyzers (FAI SWAM DC and 5A models in Milan Pascal and Limito, respectively). The PM analysers are checked for temperature, pressure, flow rates, leaks, and other operational parameters every 3 months. A

periodical comparison with gravimetric samples has been performed once yearly in Milano Pascal, and upon a specific audit programme in Limito. In 2019, annual average concentrations of NO, $NO_2$, $O_3$ were respectively 25, 37, 46 µg m$^{-3}$ in Milano Pascal, 29, 45, 46 µg m$^{-3}$ in Cormano and 26, 34, 44 µg m$^{-3}$ in Limito. For $PM_{10}$, 2019 annual averages were 29 µg m$^{-3}$ and 31 µg m$^{-3}$ in Milan Pascal and Limito, respectively. Data are available on line at www.arpalombardia.it.

**2.3 CAMS-Ensemble forecast description**

The Copernicus Atmospheric Monitoring Service (CAMS) provides, daily, 4-day ahead air quality forecasts for Europe from currently nine different regional air quality models (CHIMERE, DEHM, EMEP, EURAD-IM, GEM-AQ, LOTOS-EUROS, MATCH, MOCAGE, SILAM). Forecasts are performed independently by all the individual regional air quality systems. An ensemble (named 'CAMS-Ensemble forecast') is calculated from individual model outputs with a median approach (Marécal et al, 2015). This method provides an optimal estimate (Riccio et al, 2007) which is rather insensitive to outliers and generally yields better

estimates than the individual models (Galmarini et al., 2018). The outputs of the different individual models are interpolated on a common regular 0.1°x0.1° latitude x longitude grid (about 10 km x 10 km) over the European domain (25°W-45°E, 30°N-72°N). Hourly pollutant concentrations are calculated for altitudes raging from the 40m-thick surface layer to 5 km. Each air quality





model is based on different chemical (gas and aerosols) and physical parameterisations, but uses the same meteorological drivers as input (the ECMWF Integrated Forecasting System, IFS) and the same anthropogenic emissions data (Kuenen et al., 2014; Denier
van der Gon et al., 2015) based on 2011 emission inventories until June 2019, and on 2016 emission inventories afterwards. As the anthropogenic emissions used by the individual models did not change to account for any lockdown measure, the air quality models continued to forecast pollutants' concentrations as if the COVID-19 epidemic had not occurred in 2020.

**2.4 Data analysis.**

**2.4.1      Pollutant concentrations**

195       To determine the specific impact of lockdown measures on concentrations of air pollutants, we compared daily observations ($Obs$) with daily expected concentrations ($Exp$) for the period 17 February – 24 May 2020, which comprises the 8 lockdown weeks (D = 9 March – 3 May 2020), the 3 weeks before the beginning (A = 17 February - 8 March 2020) and the 3 weeks after the end of the lockdown period (P = 4 – 24 May 2020). NO$_2$, PM$_{10}$, NO, O$_3$ and SO$_2$ observed and expected concentrations are shown in Figure 3. Expected concentrations were derived from 2020 CAMS –Ensemble forecasts, which account for variations
in meteorological conditions and seasonal changes in emission source strengths in a "business as usual" world, i.e. without lockdown measures. However, since data from 2019 show that the agreement between CAMS-Ensemble forecasts ($CAMS_{2019}$) and observations ($Obs_{2019}$) improves from February to May (see Figure S1-S3 in Supplement), $CAMS_{2020}$ were corrected for this seasonality. Thus, 2020 daily expected pollutants concentrations $Exp$ were calculated as Eq. (1):

$$Exp = \frac{CAMS_{2020}}{CAMS_{2019}} \, Obs_{2019} \qquad\qquad\qquad\qquad\qquad\qquad\qquad\qquad\qquad\qquad (1)$$

205       The comparison of observations with these expected concentrations for 2020 has the great advantage of being insensitive to the fact that the emissions inventories used to calculate CAMS-Ensemble forecast data for 2019 and 2020 were different. The disadvantage of this approach is that $Obs$ and $Exp$ cannot be compared to each other on a daily basis, since $Exp$ values are affected by random variations in the $Obs_{2019}$/ $CAMS_{2019}$ ratio. Therefore 2020 $Obs$ and $Exp$ data were compared statistically for the 3 periods A, D, and P. Since changes in pollutant emission rates are expected to result in changes in pollutant
concentrations in terms of percentages or ratios, statistical analyses were performed on $Obs$ / $Exp$ daily ratios. We calculated occurrence frequency distributions of the $Obs$ / $Exp$ ratio using 8 class bins ranging from <0.25 to >2, all equally wide on a logarithmic scale (except the last one when specifically indicated). Cumulative frequencies of occurrence were also plotted to facilitate comparisons (Figure 4). To detect possible specific impacts of lockdown measures on highest concentrations, specific occurrence frequency distributions were also calculated by selecting the 28 days on which CAMS-Ensemble forecast data were
greater than the median during the lockdown period. These days are different for each pollutant and each site. The statistical significance of the differences in $Obs$ / $Exp$ ratios during the lockdown period in comparison with before and after the lockdown period (i.e. between *A* and *D or P* and *D*) was assessed by applying a t-test assuming unequal variances to the means $\bar{A}$, $\bar{P}$, and $\bar{D}$, defined as Eqs (2):

$$\bar{D} = mean\left(log\left(\frac{(Obs/CAMS)_{during \, lockdown}}{(Obs/CAMS)_{10 \, Mars-25 \, May \, 2019}}\right)\right), \bar{A} = mean\left(log\left(\frac{(Obs/CAMS)_{before \, lockdown}}{(Obs/CAMS)_{17 \, Feb-9 \, Mars \, 2019}}\right)\right), \bar{P} = mean\left(log\left(\frac{(Obs/CAMS)_{after \, lockdown}}{(Obs/CAMS)_{5-25 \, May \, 2019}}\right)\right) \quad (2)$$

The null hypotheses ($\bar{D} = \bar{A}$, and $\bar{D} = \bar{P}$) were tested at the 95% confidence level, and results were used to determine if differences between $\bar{D}$ and $\bar{A}$ and $\bar{D}$ and $\bar{P}$ were statistically significant.



### 2.4.2 Intensive aerosol variables

To complement our analyses based on pollutant concentrations, we also looked at two characteristics of the atmospheric aerosol measured at ECAtmO in Ispra. The first one is the percentage in number of tiny particles with mobility diameters ($D_p$) between 15 and 70 nm as compared with the "total" number of particles with mobility diameters between 15 and 800 nm. This percentage was calculated from full particle number size distributions ($10 < D_p < 800$ nm). Smallest particles ($10 < D_p < 15$ nm) were not considered because their measurement is affected by larger uncertainties (Wiedensohler et al., 2018), and by nucleation particle bursts. The range $15 < D_p < 70$ nm was selected as representative of particles emitted by primary sources (Giechaskiel et al., 2019; Giechaskiel, 2020; Ozgen et al., 2017; Tiwari et al., 2014). The second variable is the aerosol light absorption Ångström exponent (AÅE). It represents the wavelength dependence of light absorption by aerosol particles. AÅE values vary with particle sources and have commonly been used to apportion pollution particles between e.g. traffic and wood burning (Sandradewi et al, 2008). Traffic emitted particles (mainly from Diesel engines) have an AÅE close to 1, while particles from wood combustion have more variable AÅEs around 2 (Sandradewi et al, 2008). The mixture of pollution particles with primary or secondary aerosol from biogenic origin can also lead to AÅE values much greater than 1. Since both variables are insensitive to air pollution dispersion, they are much less variable than the extensive variables (i.e. atmospheric concentrations) they are derived from (e.g. Putaud et al., 2014). The values expected for these so-called intensive variables were calculated as the arithmetic averages observed during the 2017-2019 period.



*Figure 3: Observed (dots) and expected (lines) 2020 concentrations (µg m⁻³) of NO₂, PM₁₀, NO, O₃, and SO₂ in Ispra (left hand side) and Milan conurbation (right hand side). Vertical lines indicate the beginning and end of the lockdown period.*


*Figure 4: Occurrence frequency distributions of 2020 observed/expected concentration ratios (Obs / Exp) for NO₂, PM₁₀, NO, O₃ and SO₂ during the lockdown period, and during the 3 weeks before and after the lockdown period in Ispra (left) and Milan conurbation (right). Lines show cumulative frequencies of occurrence. Dashed lines show the cumulative frequency of occurrence of (Obs / Exp) ratios for the 28 days corresponding to the highest CAMS forecast values. NB: the last bin for NO in Ispra contains all values > 2.*




## 3 Results and discussion

The observation and CAMS-Ensemble forecast data used to estimate the values expected for the air pollution variables discussed in this section are described in Sections 1 and 2 of the Supplement to this article.


### 3.1. Regional background (Ispra)

The trend in AÅE observed in Ispra in 2017-2019 (and also in 2020) is consistent with a decreasing contribution of wood burning to particulate pollution from winter to summer. The AÅE values measured in 2020 can of course not be compared point to point to the 2017-2019 average in Figure 5 because the use of wood fuel for domestic heating also depends on weekend and cold evening occurrences. However, the clear increase in the AÅE average between 9 March and 4 May 2020 compared to the 3

weeks before, the 3 weeks after, and the corresponding period in 2017 - 2019 undoubtedly shows a change in particle sources related to lockdown measures (Table 1). A specific analysis focused on the 4 first weeks of the lockdown period (before significant amounts of biogenic aerosols are expected) suggest a - 45% reduction in aerosol from traffic (and a concomitant + 45% increase in aerosol from wood combustion) during that period.

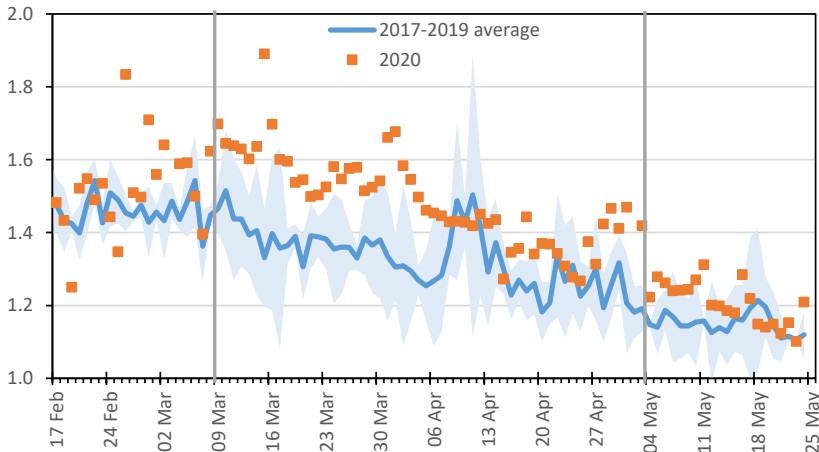

*Figure 5: Aerosol light absorption Ångström exponent (AÅE) in 2020 (dots) compared to its 2017 – 2019 average (lines). The shaded area represents ± 1 standard deviation of the average. Vertical lines indicate the beginning and the end of the lockdown period.*

Particle number size distribution measurements in Ispra typically show modes at 25 – 50 nm during morning rush hours as well as in the evening in winter. Particle primary sources include fuel combustion by thermal engines and liquid (oil) or solid fuel

(e.g. wood) combustion for domestic heating. The ultrafine mode diameters of primary particle emissions range from 50 to 100 nm for domestic heating (e.g. Tiwari et al, 2014; Ozgem et al., 2017), and range from 10 to 90 nm for engines (e.g. Giechaskiel et al, 2019; Giechaskiel, 2020). Measurements also show that peaks in the number of 15-70 nm particles can result from the growth of nucleation particles in the afternoon. The percentage of 15-70nm particles generally increased from mid-February till end of May in 2017 - 2019 (Figure 6). Considering that (1) wood burning combustion for domestic heating did not decrease during the

lockdown period, (2) nucleation and growth of secondary aerosol particles were observed on sunny days during the lockdown





period from 6 April 2020, and (3) that mostly morning peaks in particle number diminished during the lockdown period especially from 11 March to 13 April 2020, the relative "disappearance" of 15 - 70 nm particles during the lockdown period (Figure 6) can be attributed to a decrease in traffic related to lockdown measures. Since the atmospheric lifetime of 15 - 70 nm particles is 3 - 12 hours, local to regional traffic was concerned. Although it significantly increased after 4 May 2020, the percentage of 15 - 70

nm particles did not get back to the level observed before the lockdown as lockdown measures were relaxed.

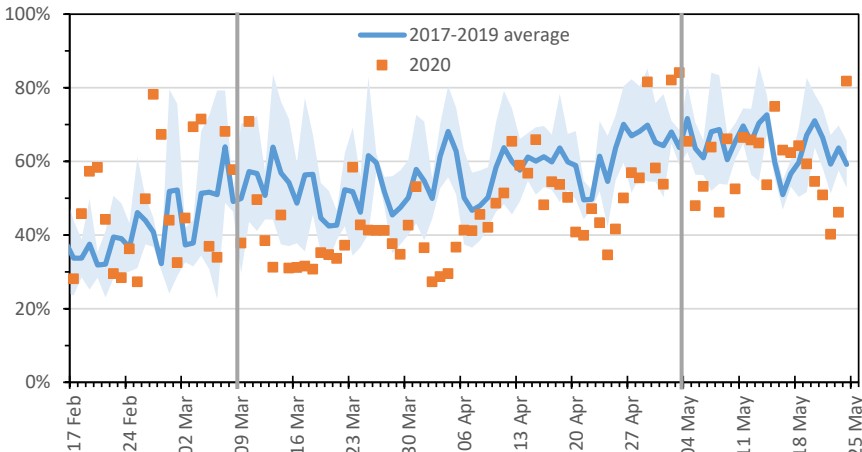

Figure 6: Percentage of sub-70 nm particles in 2020 (dots) compared to 2017 – 2019 average (line). The shaded area represents ± 1 standard deviation of the average. Vertical lines indicate the beginning and the end of the lockdown period.

The decrease in emissions from local traffic indicated by the drop in the percentage of the smallest particles (Figure 6) is the

most probable cause for the decrease of NO related to the lockdown measures in Ispra (Figure 4). NO daily mean concentrations are indeed dominated by their morning peak corresponding to traffic rush hours (which disappears during weekends). During daytime, NO is rapidly converted to $NO_2$ and NO concentrations reach very low steady state values. Decreased NO emissions should therefore result in decreases in $NO_2$. Such a decrease in $NO_2$ (-40% on average) actually occurred in Ispra as a result of the lockdown measures from 9 March 2020 (Figure 4). In contrast, $NO_2$ did not "recover" from the lockdown measures, unlike NO of

which concentrations increased again in comparison to expected concentrations as lockdown measures were relaxed on 4 May 2020 (Table 1). Due to its lifetime of about 1-2 days (Seinfeld and Pandis, 2016), $NO_2$ can travel rather long distances. Nitrogen oxides are also emitted by large combustion sources like thermal power plants, which also emit $SO_2$. However, our analysis of $SO_2$ data also reveals that sources of $SO_2$ that affect concentrations in Ispra decreased due to lockdown measures (Figures 3 and 4). The fact that $NO_2$ observed / expected ratios remained as low after, as during the lockdown period, could be explained by a

slower increase in traffic on the regional scale as compared to the local scale.



*Table 1: Observed / expected ratios for pollutant concentrations and aerosol characteristics before, during and after the lockdown measures in Ispra (regional background) and Milan (urban background).*

|  | Ispra (Regional Background) | | | | Milan (Urban Background) | | | |
| --- | --- | --- | --- | --- | --- | --- | --- | --- |
|  | before | during | during (> median) | after | before | during | during (> median) | after |
| $NO_2$ | 1.52 | 0.89 | 0.78 | 0.82 | 0.83 | 0.57 | 0.59 | 0.73 |
| $PM_{10}$ | 0.67 | 0.60 | 0.61 | 0.77 | 0.77 | 0.83 | 0.81 | 1.20 |
| NO | 1.95 | 1.11 | 1.09 | 1.60 | 0.69 | 0.70 | 0.58 | 0.99 |
| $O_3$ | 0.79 | 0.87 | 0.96 | 0.98 | 0.93 | 1.17 | 1.22 | 1.15 |
| $SO_2$ | 1.19 | 1.02 | 0.85 | 1.59 |  |  |  |  |
| sub 70nm % | 1.09 | 0.79 | 0.75 | 0.91 |  |  |  |  |
| AÅE | 1.04 | 1.12 | 1.14 | 1.05 |  |  |  |  |

Regarding secondary pollutants, the highest $O_3$ concentrations significantly increased compared to expected concentrations during the lockdown period in comparison with the 3 weeks before (Figure 7 and 4). This suggests that $O_3$ peaks are usually diminished by NO titration during this period of the year in Ispra, and that the abatement in $NO_x$ emissions revealed by NO and $NO_2$ data analyses led to a reduction of this effect. The relaxation of lockdown measures led to a further increase in $O_3$. Since $O_3$ production is generally VOC limited in May in Ispra, this increase in $O_3$ is probably due to an increase in anthropogenic emissions of VOCs from e.g. local traffic. In the case of $PM_{10}$, which is mainly composed of secondary particulate species in Ispra (Larsen et al., 2012), no significantly decrease compared to expected concentrations could be identified as lockdown measures were implemented (Figure 3 and 4). This is because the decrease in $PM_{10}$ related to traffic was compensated by the increase from wood burning for domestic heating, at least during the first half of the lockdown period. In contrast, $PM_{10}$ significantly but marginally increased as lockdown measures were relaxed on 4 May 2020 at a time of the year (from May onwards) where wood burning combustion for domestic heating is largely reduced.

### 3.2. Urban background (Milan conurbation)

To represent Milan urban background, we used data from 3 urban background sites located in Milan hinterland and in Milan city centre (Figure 1). $NO_2$ significantly decreased (-30% on average) compared to expected concentrations as lockdown measures were implemented (Figures 3 and 4). $NO_2$ significantly but not totally "recovered" when lockdown measures were relaxed (Table 1), which suggests that not all sources determining $NO_2$ concentrations in Milan were fully reopened. However, the increase in NO after the end of the lockdown period suggests that local traffic largely resumed. Perhaps NOx emissions on a broader scale did not yet reach their usual intensity during the 3 first weeks after the end of the lockdown period, as already suggested by $NO_2$ data from Ispra. Regarding NO, it should be noticed that a significant decrease in comparison to the weeks before the lockdown period could only be detected at the city centre station (Figure S8). Both sites in the hinterland are much closer to highways and might reflect more NO emissions from heavy duty vehicles, whose traffic did not decrease that much at least during the first weeks of the lockdown period.

As in Ispra, the implementation of lockdown measures on 9 March 2020 led to increases of $O_3$ in Milan conurbation compared to expected concentrations (Figures 3 and 4). This can be explained by the decrease of $O_3$ titration by NO in a pollution regime where photochemical $O_3$ production is limited by the availability of volatile organic compounds. The relaxation of lockdown measures did not lead to the expected decrease in $O_3$ (Figure 4), perhaps because NOx emissions did not fully recover during the 3 weeks following the end of the lockdown period.





Again, as in Ispra, no significant change in $PM_{10}$ could be detected when lockdown measures were implemented. This is very likely due to the fact that decreased emissions of $PM_{10}$ (and $PM_{10}$ precursors) were compensated by increases from other sources like domestic heating and enhanced formation of secondary PM. In particular, Huang et al. (2020) reported that increased oxidative capacities of the atmosphere (e.g. higher $O_3$ concentrations) resulted from the drastic reductions in $NO_x$ emissions

following from the lockdown measures in China, that in turn lead to an increase in the formation rate of nitric acid ($HNO_3$). Such a phenomenon in northern Italy, together with sustained emissions of ammonia from agriculture, which was not affected by the lockdown (ARPA Lombardia), could have resulted in increased formation of particulate ammonium nitrate ($NH_4NO_3$) and therefore an increase in $PM_{10}$ concentrations beyond expected concentrations in the Milan conurbation. For the 3 weeks following 4 May 2020, the relaxation of lockdown measures led to a further increase in $PM_{10}$ in comparison to expected

concentrations in Milan. This might be attributed to the upturn in traffic and particularly to the re-suspension of dust from roads that had been little used for several weeks.

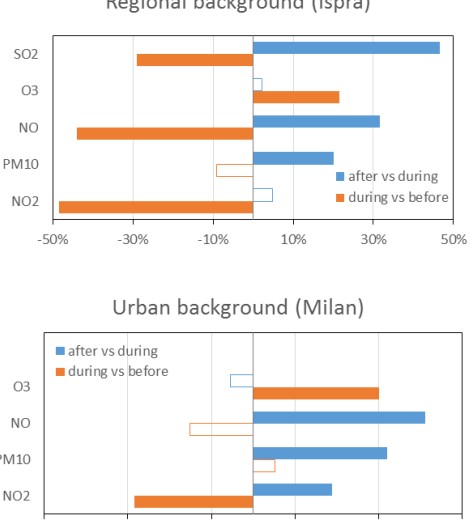

*Figure 7: Changes in observed / expected concentration for the 3 weeks before and after the lockdown in comparison with the 28 days of lockdown corresponding to 50% highest CAMS daily forecasts for each pollutant. Filled bars represent statistically*

*significant differences. Empty bars represent differences that are not significantly different from zero.*

## 4     Conclusions

Northern Italy has been an air pollution hot spot for decades due to a high population density, intense economic activities, and particular orographic and meteorological conditions. Northern Italy also hosted the very first clusters of COVID-19 epidemic in Europe from February 2020, while containment measures were gradually implemented from the end of February with strict

lockdown measures in force from 9 March to 4 May 2020. Lockdown measures impacted numerous economic sectors and activities with potential impacts on air pollution. Concentrations of several pollutants decreased from February to May 2020, as they have done every year for several decades, mainly due to seasonal variations in emissions and meteorological conditions. We





isolated the specific impact of lockdown measures on air pollution by comparing observed to expected data at one regional background site (Ispra) and 3 urban background sites (in Milan conurbation) for the period 17 February -24 May 2020. All 4

stations were in the COVID-19 "red zone". Expected pollutant concentrations were derived from CAMS Ensemble forecasts, which are based on actual meteorological conditions and historical emissions estimates that ignored the COVID-19 epidemic and related lockdown measures. Changes in observed pollutant concentrations compared with computed expected concentrations for the lockdown period and the 3 weeks before and after the lockdown period therefore directly reflect the impact of lockdown measures on air pollution.

We showed that lockdown measures had statistically significant impacts on concentrations of most gaseous pollutants (Table 1). However, we were not able to highlight systematic effects on $PM_{10}$ concentrations, probably due to the wide variety of primary and secondary sources of $PM_{10}$.

Focusing on the effect of the lockdown measures on days for which the CAMS ensemble model forecast concentrations were above the median for the lockdown period (Figure 7), we found that $NO_2$ concentrations decreased by about -30% and -

50% at the urban and regional background sites, respectively, due to the lockdown implementation on 9 March 2020. This is consistent with similar decreases in NO concentrations. The relaxation of lockdown measures on 4 May led to a partial recovery in $NO_2$ concentrations in Milan (urban background), but not in Ispra (regional background). Unlike $NO_2$, $PM_{10}$ concentrations were not significantly affected by lockdown measures. The increase in the aerosol light absorption Ångström exponent observed in Ispra during the lockdown period in comparison with the same period in 2017 – 2019 indicates that this is because the decrease

in traffic-related $PM_{10}$ was compensated by an increase in $PM_{10}$ associated with wood burning for domestic heating. $PM_{10}$ concentrations in Milan are to a great extent influenced by $PM_{10}$ 'non urban' and 'non traffic' sources (Thunis et al., 2018), including the formation of secondary aerosol. Sustained regional background $PM_{10}$ concentrations and a modified $HNO_3$ production regime associated with continuing $NH_3$ emissions from agriculture could explain the lack of decrease in $PM_{10}$ resulting from the implementation of the lockdown measures in Milan too. In contrast, the relaxation of lockdown measures led to an

increase of $PM_{10}$ concentrations at both urban and regional background sites (+20% and + 15%, respectively) in May, when domestic heating is much reduced. Specific aerosol data from Ispra suggest that the impact of both local pollution sources (traced by the percentage of particles with diameters between 15 and 70 nm) and major industrial pollution sources (traced by $SO_2$ concentrations) on regional background air pollution decreased by -30 to -40% when lockdown measures were implemented, and at least partially got back to "normal" when lockdown measures were relaxed. Lastly, it can be pointed out that lockdown

measures led to an increase in the highest $O_3$ concentrations at both the urban and regional background sites when they were implemented.

The sad experience of the COVID-19 epidemic and subsequent lockdown measures shows that drastic changes in mobility and economic activity can lead to 0% (insignificant) to -30 % reductions in air pollution in urban background areas. These figures suggest that the abatement of air pollution down to levels that do not have adverse effects on human health in northern Italy

may require structural changes in energy production, domestic heating, agriculture and transport.

**Data availability.** Observation data from Ispra are available at https://data.jrc.ec.europa.eu/collection/abcis and https://actris.nilu.no/. Observation data from Milan are available at https://www.arpalombardia.it/Pages/Aria/Richiesta-Dati.aspx. Model forecast data for all sites are available at https://ads.atmosphere.copernicus.eu/cdsapp#!/dataset/cams-
europe-air-quality-forecasts?tab=form



**Author contributions.** Contributed to conception and design: JPP, EP, LP. Contributed to acquisition of data: SMDS, FL, UDS. Contributed to analysis and interpretation of data: JPP, EP, LP, GL, AC. Drafted the article: JPP, EP, LP, GL, AC.

**Competing interests.** The authors declare that they have no conflict of interest.

**Acknowledgements.** This work was partially supported by the European Union's projects ACTRIS-2 (EU Horizon 2020 – No 654109)
and ACTRIS-Implementation (EU Horizon 2020 - No 871115). The authors thank the Copernicus Atmosphere Monitoring Service Information, in particular the Regional Production Service. JPP, LP, EP, SMDS, and FL thank their colleagues from JRC for fruitful tele-discussions during the whole lockdown period and for helpful comments on the manuscript.



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
