# Peer review of "Impacts of the COVID-19 lockdown on air pollution at regional and urban background sites in northern Italy."

_Atmospheric Chemistry and Physics, 2020_

## Referee Comment (RC1) · Anonymous Referee #2 · 4 Nov 2020

The manuscript provides details of the impacts of COVID-19 lockdown measures on air quality in northern Italy. The results are supported by observations from (1) the European Commission atmospheric observatory of Ispra and (2) regional environmental protection agency (ARPA) air monitoring stations in Milan, and (3) CAMS model simulations. The topic of the manuscript is suitable for the ACP. The manuscript is well written. I suggest the authors including the below-mentioned comments in the revised version.

(1)Introduction section: A Pi-diagram of sectoral emissions in Italy will be useful.

(2) CAMS-Ensemble forecast description: This section should be elaborated. Details

of the accuracy of forecast in terms of statistical significance should also be stated here, although some description is given in section 2.4.1.

(3) A comparison of the percentage change in pollutants in Italy with other countries, during the lockdown should be provided using the literature.

(4) Line 19: Do you mean surface concentrations of atmospheric pollutants ($NO_2$, $PM10$, $O_3$, NO,.........

(5) Line 80: tropospheric $NO_2$ column concentration from Sentinel-5p or surface $NO_2$? Please clarify.

(6) Is there an influence of dust aerosol during the lockdown period which is seen in $PM10$ observations?

(7) The reason for using "the same anthropogenic emissions data based on 2011 emission inventories until June 2019, and on 2016 emission inventories afterward" should be stated for clarity purpose. The forecast is highly modulated by the emission inventories.

(8) Figure 3: I suggest plotting standard deviation on observed data. Whether the expected values fall within the range of standard deviation?

(9) To strengthen section 3.1, I suggest providing a plot of the distribution of the VOC/NOX ratio and related discussions.

(10) Also, discuss if the urban and regional background sites in Italy have experienced ozone enhancement (reduced titration of O3 by NO) in the past? Is there any specific feature during lock-down?

---

## Author Comment (AC1) · 25 Nov 2020

We would like to thank Referee #2 for constructive comments to our manuscript posted on 4 November, and we are glad to discuss those interactively.

1. We include below a pie-plot showing NOx and VOC 2015 sectorial emissions in Italy as derived from the EDGAR emission inventory. We would be glad to know if Referee #2 thinks that a pie-plot showing SO2 emissions would also be useful. These pie plots will be included in the Supplement to our manuscript.

2. Regarding CAMS-Ensemble forecast description, we intended to include a para-

graph as follows: "The CAMS Regional Air Quality forecast are routinely quality controlled and dedicated evaluation report are published on a quarterly basis for both individual and ENSEMBLE model. The relevant report for March/April/May 2020 is available at https://atmosphere.copernicus.eu/sites/default/files/2020-09/CAMS50_2018SC2_D5.2-3.1.ENSEMBLE-2020MAM_202008_NRTProduction_Report_v1.pdf. For the specific case of PM10, the RMSE ranged from 6.8 and 9.1 $\mu$g/m$^3$ for D+0 of the forecast, which is well below the key performance indicator defined at 18 $\mu$g/m$^3$. The corresponding bias is between -0.3 and -0.1 $\mu$g/m3, and spatial correlation ranges from 0.71 to 0.75. The precise definition of those statistical scores is available at https://regional.atmosphere.copernicus.eu/doc/USER_GUIDE_VERIFICATION_STATISTICS.pdf. We will add a table with such statistics for all pollutants in the Supplement of the revised manuscript. However, we would like to underline that our study is based on the statistical analysis of the changes between 3 periods (before, during, and after the lockdown) in the differences "observed concentrations - expected concentrations" rather than on the comparison between observations and forecasts themselves.

3. There are not many published peer reviewed papers on the effect of lockdown measures on air pollution yet. We list changes observed in Brazil, China, and France in the introduction. Since the MS was submitted, a couple of studies were published for Spain (Madrid, Barcelona) and Europe as a whole (doi.org/10.1016/j.scitotenv.2020.140353, doi.org/10.5194/acp-2020-995, doi.org/10.5194/acp-2020-1171), the results of which will be mentioned in the revised version of our manuscript.

4. Line 19: we will specify in the abstract that we deal with ground-level concentrations.

5. Line 80 specifies that "total column-integrated NO2 tropospheric concentrations over northern Italy" were derived "using Sentinel-5p data".

6. There was actually a desert dust outbreak reaching the measurement locations on March 28-29th (during the lockdown). This outbreak affected 2 days amongst 56 (8 weeks). We will compute statistics excluding these 2 days and state in the MS if this

leads to any significant difference or not.

7. CAMS-Ensemble forecast made use of "anthropogenic emissions data based on 2011 emission inventories until June 2019, and on 2016 emission inventories afterwards" to stay up-to-date. This was not our own choice. 2019 forecast and observed data are used only to correct for seasonal changes in the difference between forecasts and observations. The way we calculate expected concentrations is independent from the change in emission inventories used by the CAMS-Ensemble model between 2019 and 2020.

8. The observation data from Ispra are 24-hr averages of measurements performed at a single location. Adding standard deviations would not be relevant. The observation data from Milan are averages from 3 urban background sites for all variables but PM10 (2), and standard deviations would not make sense. We would be glad to know if Referee #2 retains useful to include error bars showing min and max values among the 3 (2) sites.

9. Unfortunately, there is no VOC data available for Ispra from March 8th to July 2nd, 2020, because only fully automatic measurements could be carried out during the lockdown. Additional information based on data from previous years will be included in the manuscript to assess the O3 production chemical regime at this location.

10. Such an experiment of dramatically reducing NOx emissions in the whole of northern Italy never occurred before. Indeed, traffic limitation measures aiming at reducing wintertime particulate pollution peak at most affect half of the vehicles, and urban areas only. It is therefore difficult to compare what happened during the COVID lockdown with any previous situation.
* * *
[Figure]

**Fig. 1.**

---

## Referee Comment (RC2) · Anonymous Referee #1 · 5 Jan 2021

The manuscript addresses the variation in air quality due to the COVID-19 restrictions in northeastern Italy. The paper is interesting, as it presents pollutants concentrations before, during, and after lockdown.

I suggest a minor revision:

1. Please improve the quality of Figure 1.

2. Figures 3 and 4 are presented before the results section, however, they only appear in the text after Figure 6. Table 1 is very distant from where it is mentioned in the text. Figure 3 is mentioned in the text after Figure 4. Please reorganize these items.

[Figure]

3. I suggest presenting the results of Figure 7 in a table.

4. I suggest writing the conclusion in a more concise way, it may be presented in topics.

---

## Author Response (AR1)

**Impacts of the COVID-19 lockdown on air pollution at regional and urban background sites in northern Italy,**

acp-2020-755, by J.-P. Putaud et al.

Detailed point-by-point response to referees' comments and specific changes in the revised manuscript.

We would like to thank both Referees for their comments to our manuscript. We are confident that they all have been properly addressed, as detailed below:

Referee #1's comments:

1.  Please improve the quality of Figure 1.

R: We improved Figure 1: same fonts and symbols for the measurement sizes on both panels, more specific/concise legend (see below).

[Figure]

2.  Figures 3 and 4 are presented before the results section, however, they only appear in the text after Figure 6. Table 1 is very distant from where it is mentioned in the text. Figure 3 is mentioned in the text after Figure 4. Please reorganize these items.

R: Figures 3 and 4 are first mentioned in Section 2.4.1 line 199 and 213, respectively. Table 1 is mentioned in Section 3.1, lines 256 (p. 11) and 286 (p. 12), and was presented in the initial version of the manuscript in line 291 (p. 13). The positioning of Table 1 in the final version of the paper will be decided by the publisher.

3.  I suggest presenting the results of Figure 7 in a table.

R: We had Figure 7 data shown in a Table in an early version of our manuscript (see Table R1 below). We deemed it was difficult to get at a glance a clear overview of our results from such a table. As a compromise, we added data labels in Figure 7 (for statistically significant changes only) in the revised version of our manuscript (see Figure 7 below).

*Table R1: Changes in observed / expected concentrations for the 3 weeks before and after the lockdown in comparison with the 28 days of lockdown for the 50% highest CAMS daily forecasts for each pollutant. Numbers in grey indicate changes that are not significantly different from zero.*

|  | during vs before Ispra | during vs before Milan | after vs during Ispra | after vs during Milan |
|---|---|---|---|---|
| NO2 | -49% | -28% | +5% | +19% |
| PM10 | -9% | +5% | +20% | +32% |
| NO | -44% | -15% | +32% | +41% |
| O3 | +21% | +30% | +2% | -5% |
| SO2 | -29% |  | +47% |  |
| sub 70nm part. % | -37% |  | +16% |  |
| AÅE | +7% |  | -7% |  |

[Figure]

[Figure]

*Figure 7: Changes in observed / expected concentrations for the 3 weeks before and after the lockdown in comparison with the 28 days of lockdown corresponding to 50% highest CAMS daily forecasts for each pollutant. Filled bars represent statistically significant differences. Empty bars represent differences that are not significantly different from zero.*

4. I suggest writing the conclusion in a more concise way, it may be presented in topics.

R: We simplified and shortened the conclusion, and split the conclusion in various topics as suggested. Please find the new version below:

*Northern Italy has been an air pollution hot spot for decades. Northern Italy also hosted the very first clusters of COVID-19 epidemic in Europe and from February 2020, containment measures were gradually implemented culminating in strict lockdown measures in force between 9 March and 4 May 2020. We isolated specific impacts of the lockdown measures on air pollution by comparing observed with expected data at one regional background site (Ispra) and 3 urban background sites (in Milan conurbation) across the period 17 February - 24 May 2020. All 4 stations were in the COVID-19 "red zone". Expected pollutant concentrations were derived from CAMS Ensemble forecasts, which are based on actual meteorological conditions and historical emissions estimates that ignored the COVID-19 epidemic and related lockdown measures. Changes in observed versus computed expected concentrations for the lockdown period and the 3 weeks before and after the lockdown period should therefore directly reflect the impact of lockdown measures on air pollution.*

*We showed that lockdown measures had statistically significant impacts on concentrations of most gaseous pollutants (Table 1). However, we were not able to highlight systematic significant effects on PM$_{10}$ concentrations.*

*Focusing on those days for which the CAMS ensemble model forecast concentrations were above the median for the lockdown period (Figure 7):*
*− NO$_2$ concentrations decreased by about -30% and -50% at the urban and regional background sites, respectively, as a result of the lockdown implementation on 9 March 2020. The relaxation of lockdown measures on 4 May led to a partial recovery in NO$_2$ concentrations in Milan (urban background), but not in Ispra (regional background);*
*− Unlike NO$_2$, PM$_{10}$ concentrations were not significantly affected by the lockdown measures. We showed that the decrease in traffic-related PM$_{10}$ was compensated by an increase in PM$_{10}$ associated with wood burning for domestic heating in Ispra.*

$PM_{10}$ concentrations in Milan are to a great extent influenced by $PM_{10}$ 'non urban' and 'non traffic' sources (Thunis et al., 2018), including the formation of secondary aerosol. Sustained regional background $PM_{10}$ concentrations and a modified $HNO_3$ production regime associated with continuing $NH_3$ emissions from agriculture could explain the lack of decrease in $PM_{10}$ resulting from the lockdown measures in Milan too. In contrast, the relaxation of lockdown measures led to an increase of $PM_{10}$ concentrations at both urban and regional background sites (+30% and + 20%, respectively) in May, when domestic heating is much reduced;

− The lockdown measures led to an increase in the highest $O_3$ concentrations at both the urban and regional background sites.

The sad experience of the COVID-19 epidemic and subsequent lockdown measures shows that drastic changes in mobility and economic activity can lead to 0% (insignificant) to -30 % reductions in air pollution in urban background areas. These figures suggest that the abatement of air pollution down to levels that do not have adverse effects on human health in northern Italy may require structural changes in other sectors including energy production, domestic heating, agriculture, in addition to transport.

Referee #2's comments:

1. Introduction section: A Pi-diagram of sectoral emissions in Italy will be useful.

R: Fig. S1 as below has been added to the Supplement.

[Figure]

2. CAMS-Ensemble forecast description: This section should be elaborated. Details of the accuracy of forecast in terms of statistical significance should also be stated here, although some description is given in section 2.4.1.

R: This section has been further elaborated. Details on the accuracy of the CAMS-Ensemble forecasts have been included. Since statistical scores are not available for all variables, we modified this section (lines 194 - 201) as copied below. Please note that CAMS-Ensemble forecasts and observations for both 2019 and 2020 for both sites and all variables are compared in the Supplement.

CAMS Regional Air Quality forecasts are routinely quality controlled and dedicated evaluation reports are published every third month for both individual and the ENSEMBLE models (see atmosphere.copernicus.eu/regional-services). In this work, we used daily averages of the CAMS-Ensemble surface concentrations forecast each day for the next 24 hours (D0). For the period March – May 2019, the difference between daily mean D0 forecasts and measurements performed at various reference stations across northern Italy (expressed as median of the root mean square errors, RMSE) were 10.5, 10.6 and 24.5 µg/m³ for $NO_2$, $PM_{10}$ and $O_3$, respectively. Additional statistical scores are available in quarterly CAMS reports (CAMS, 2019; CAMS, 2020c). Note that the actual CAMS-Ensemble RMSEs relative to the stations and time periods we analyzed are part of our statistical analysis described in Section 2.4.1.

3. A comparison of the percentage change in pollutants in Italy with other countries, during the lockdown should be provided using the literature.

R: There were not yet many published peer reviewed papers on the effect of lockdown measures on air pollution when we submitted our manuscript in July 2020. We listed changes observed in Brazil, China, and France in the introduction. Since the MS was submitted, a couple of studies were published for Spain (Madrid, Barcelona) and Europe as a whole (doi.org/10.1016/j.scitotenv.2020.140353, doi.org/10.5194/acp-2020-995, doi.org/10.5194/acp-2020-1171), the results of which are mentioned in the revised version of our manuscript (line 76-78) as below:

*Across Europe, Grange et al. (2020) estimated that $NO_2$ and $O_3$ concentrations at urban background sites were 32 % lower and 21 % higher than expected, respectively, when maximum mobility restrictions were in place. A clear decrease of $NO_2$ concentrations in Barcelona and Madrid (Spain) during the lockdown was also described by Baldasano (2020).*

4.   Line 19: Do you mean surface concentrations of atmospheric pollutants (NO2, PM10, O3, NO,…).

R: We specified in the abstract (line 19) that we deal with ground-level concentrations:

*On the other hand, ground-level measured concentrations of atmospheric pollutants ($NO_2$, $PM_{10}$, $O_3$, NO, $SO_2$) were compared to expected concentrations …*

5.   Line 80: tropospheric NO2 column concentration from Sentinel-5p or surface NO2? Please clarify.

R: Line 80 (currently 83) specified "… *maps of $NO_2$ surface concentrations estimated from satellite data…*".

6.   Is there an influence of dust aerosol during the lockdown period which is seen in PM10 observations?

R: There was actually a desert dust outbreak reaching the measurement locations on March 28-29[th] (during the lockdown). This outbreak affected 2 days amongst 56 (8 weeks). Statistics were computed excluding these 2 days and results did not significantly differ. A sentence was added in the revised manuscript (lines 261-264) as below:

*The high $PM_{10}$ concentrations observed at all sites on 28 and 29 March 2020 were related to desert dust advection from the east (see maps from the Sand and Dust Storm Warning Advisory and Assessment System at the WMO SDS-WAS web site). The data from these two days were not excluded from our statistical analysis since they did not affect its results.*

7.   The reason for using "the same anthropogenic emissions data based on 2011 emission inventories until June 2019, and on 2016 emission inventories afterward" should be stated for clarity purpose. The forecast is highly modulated by the emission inventories.

R: CAMS-Ensemble forecast made use of "*anthropogenic emissions data based on 2011 emission inventories until June 2019, and on 2016 emission inventories afterwards*" to stay up-to-date. This was not our own choice. 2019 forecast and observed data are used only to correct for seasonal changes in the difference between forecasts and observations. The way we calculate expected concentrations is independent from the change in emission inventories used by the CAMS-Ensemble model between 2019 and 2020.

8.   Figure 3: I suggest plotting standard deviation on observed data. Whether the expected values fall within the range of standard deviation?

R: The observation data from Ispra are 24-hr averages of measurements performed at a single location. Adding standard deviations would not be relevant. The observation data from Milan are averages from 3 urban background sites for all variables but PM10 (2), and standard deviations would not make much sense.  Our statistical analyses actually determine if observed concentrations were "on average" different from expected values.

9.   To strengthen section 3.1, I suggest providing a plot of the distribution of the VOC/NOX ratio and related discussions.

R: Unfortunately, there are no VOC data available for Ispra between March 8[th] and July 2[nd], 2020, because only fully automatic measurements could be carried out at that site during the lockdown. Additional information based on data from previous years were included in the initial version of the manuscript (lines 130-132, currently 133-135) to assess the $O_3$ production chemical regime at this location, as follows:

*Past measurements of HCHO/$NO_2$ ratios compared to the threshold values proposed by Tonnensen and Dennis (2000) suggest that the photochemical production of $O_3$ is limited by the availability of volatile organic compounds in February – May in Ispra.*

10. Also, discuss if the urban and regional background sites in Italy have experienced ozone enhancement (reduced titration of O3 by NO) in the past? Is there any specific feature during lock-down?

R: Such an experiment of dramatically reducing NOx emissions in the whole of northern Italy never occurred before. Indeed, traffic limitation measures aiming at reducing wintertime particulate pollution peak affect at most half of the vehicles, and urban areas only. It is therefore difficult to compare what happened during the COVID lockdown with any previous situation. A modelling work addressing the impact of NOx emission reductions on PM concentrations in northern Italy (acp-2021-65 submitted on 22 January 2021) also confirms that O3 is expected to increase as NOx emissions decrease in NOx-rich areas.